# Loss of Aryl Hydrocarbon Receptor Favors *K-Ras*^G12D^-Driven Non-Small Cell Lung Cancer

**DOI:** 10.3390/cancers13164071

**Published:** 2021-08-13

**Authors:** Ana Nacarino-Palma, Claudia M. Rejano-Gordillo, Francisco J. González-Rico, Ana Ordiales-Talavero, Ángel C. Román, Myriam Cuadrado, Xosé R. Bustelo, Jaime M. Merino, Pedro M. Fernández-Salguero

**Affiliations:** 1Departamento de Bioquímica y Biología Molecular y Genética, Facultad de Ciencias, Universidad de Extremadura, Avenida de Elvas s/n, 06071 Badajoz, Spain; ana.nacarino.p@gmail.com (A.N.-P.); claudiamrg@unex.es (C.M.R.-G.); fjgonzalez@unex.es (F.J.G.-R.); aordialest@unex.es (A.O.-T.); acroman@unex.es (Á.C.R.); jmmerino@unex.es (J.M.M.); 2Instituto Universitario de Investigación Biosanitaria de Extremadura (INUBE), Avenida de la Investigación s/n, 06071 Badajoz, Spain; 3Mechanisms of Cancer Program, Centro de Investigación del Cáncer, Campus Unamuno s/n, 37007 Salamanca, Spain; mcuadrado@usal.es (M.C.); xbustelo@usal.es (X.R.B.); 4Instituto de Biología Molecular y Celular del Cáncer, CSIC-University of Salamanca, Campus Unamuno s/n, 37007 Salamanca, Spain; 5Centro de Investigación Biomédica en Red de Cáncer (CIBERONC), Campus Unamuno s/n, 37007 Salamanca, Spain

**Keywords:** aryl hydrocarbon receptor, *K-Ras^G12D^*, lung carcinogenesis, lung stem cells, NSCLC, organoids

## Abstract

**Simple Summary:**

Non-small cell lung cancer (NSCLC) accounts for over 80% of the total number of lung cancers, thus having an important impact on human health worldwide. Despite the identification of different mutations in NSCLC patients, this cancer type has limited therapeutic approaches in part due to the existence of diverse mutational profiles among patients that challenges the application of effective treatments. The glycine to aspartic mutation (G to D) in the *K-Ras* oncogene (G12D) is a common genetic alteration considered relevant at the initial stages of NSCLC. The aryl hydrocarbon receptor Ahr, on the other hand, can either suppress or promote tumor development depending on the cell phenotype. We report here that Ahr expression can limit *K-Ras^G12D^*-induced NSCLC in vivo and that it does so by controlling the expansion of lung stem cells expressing pluripotency markers. Non-toxic physiological ligands for Ahr may be good tools to reduce the NSCLC burden in cases of *K-Ras^G12D^* activation.

**Abstract:**

Non-small cell lung adenocarcinoma (NSCLC) bearing *K-Ras^G12D^* mutations is one of the most prevalent types of lung cancer worldwide. Aryl hydrocarbon receptor (AHR) expression varies in human lung tumors and has been associated with either increased or reduced lung metastasis. In the mouse, Ahr also adjusts lung regeneration upon injury by limiting the expansion of resident stem cells. Here, we show that the loss of Ahr enhances *K-Ras*^G12D^-driven NSCLC in mice through the amplification of stem cell subpopulations. Consistent with this, we show that *K-Ras^G12D^;Ahr^−/−^* lungs contain larger numbers of cells expressing markers for both progenitor Clara (SCGB1A1 and CC10) and alveolar type-II (SFTPC) cells when compared to *K-Ras^G12D^;Ahr^+/+^*-driven tumors. They also have elevated numbers of cells positive for pluripotent stem cells markers such as SOX2, ALDH1, EPCAM, LGR5 and PORCN. Typical pluripotency genes *Nanog, Sox2* and *c-Myc* were also upregulated in *K-Ras^G12D^;Ahr^−/−^* lung tumors as found by RNAseq analysis. In line with this, purified *K-Ras^G12D/+^;Ahr^−^/^−^* lung cells generate larger numbers of organoids in culture that can subsequently differentiate into bronchioalveolar structures enriched in both pluripotency and stemness genes. Collectively, these data indicate that Ahr antagonizes *K-Ras*^G12D^-driven NSCLC by restricting the number of cancer-initiating stem cells. They also suggest that Ahr expression might represent a good prognostic marker to determine the progression *of K-Ras^G12D^*-positive NSCLC patients.

## 1. Introduction

Lung cancer is responsible for several millions deaths worldwide each year. Non-small cell lung cancer (NSCLC) is the predominant subtype accounting for nearly 85% of patients, with adenocarcinomas and squamous cell carcinomas representing 50% and 40% of total cases, respectively [1,2]. Among mutations described in human lung tumors, those in the Epidermal Growth Factor Receptor (EGFR), Anaplastic Lymphoma Kinase (ALK), Human Epidermal Growth Factor Receptor 2 (HER2) and the Small GTPase *K-Ras* are frequently found in patients [3,4,5,6]. *K-Ras* appears to the activated at the initial stages of lung tumorigenesis [7] and may represent a suitable biomarker to predict tumor progression and treatment response [6,8]. Nonetheless, its heterogeneous mutational pattern [4] and its poor druggable nature [9] complicate its therapeutic value.

Large-scale studies including the Lung Cancer Mutation Consortium indicate that *K-Ras* mutations affect about one-third of human lung adenocarcinomas [10]. Additional work has also revealed that single nucleotide *K-Ras* mutations [11], involving amino acids G12, G13 and Q61, lead to reduced GTPase activity and constitutive oncogene activation [12]. Particularly, glycine 12 conversion to aspartic acid (G12D) is a frequent event in human NSCLC, conferring a poor prognosis to patients [13] and inducing the formation of multifocal clonal adenomas in the mouse [14]. It is known that *K-Ras*^G12D^ interacts with several signaling networks in lung adenocarcinoma. Thus, Hippo pathway effector YAP appears needed for *K-Ras*-induced lung cancer [15], whereas the *K-Ras*^G12D^/EGFR axis sustains AT2 cells’ stemness during lung transformation [14]. Other proteins such as NK2 homeobox-1 (Nkx2-1) can also enhance the ability of *K-Ras*^G12D^ to induce pulmonary tumors. Haploinsufficiency of Nkx2-1 enhanced *K-Ras*^G12D^-mediated tumor progression [16]. Considering the very limited benefit offered by therapies targeting *K-Ras*, it is important to identify novel molecular interactors that may help design novel clinical approaches to improve the prognosis and therapy of *K-Ras*-dependent lung cancers. The *K-Ras*^G12D^ conditional targeted allele (*LSL-K-Ras*^G12D^) can be activated in vivo in mice by administration of a Cre recombinase-expressing adenovirus (AdenoCre). When absorbed through the nasal epithelium, AdenoCre can initiate tumorigenic processes in the lung [7].

The aryl hydrocarbon receptor (Ahr) is a transcription factor with known roles in tumor progression and dissemination in different tissues and organs [17,18,19,20]. Previous work has revealed that Ahr affects tumor development and spreading in a cell and tissue-type dependent manner. Ahr overexpression has been associated with human NSCLC by sustaining cell stemness through Jak/Stat3 [21], by mediating the toxic effects of benzo-[*a*]-pyrene [22] and through the upregulation of metabolizing enzymes such as CYP1B1 [23,24]. On the contrary, Ahr has tumor suppressor activity in NSCLC since its knockdown induces epithelial-to-mesenchymal transition and invasiveness of lung tumor cells [25] while its upregulation activates the TGFβ-Smad2 pathway, suppressing lung metastasis [26]. Moreover, Ahr deficiency enhances lung regeneration in the mouse through the expansion of undifferentiated Clara and Basal cells and of epithelial cells expressing pluripotency markers Nanog and Oct4 [27].

In this work, we have investigated whether Ahr cooperates or interferes with oncogenic *K-Ras*^G12D^ in NSCLC in vivo. Our main conclusion is that Ahr deficiency stresses *K-Ras*^G12D^-induced NSCLC, likely by amplifying undifferentiated and pluripotent epithelial cell populations, eventually increasing tumor growth. Therefore, Ahr has tumor suppressor activity in the lung limiting the oncogenic potential of *K-Ras*^G12D^. Physiological, non-toxic Ahr ligands may represent potential tools to attenuate the oncogenic activity of *K-Ras* in the lung.

## 2. Materials and Methods

### 2.1. Ahr and K-Ras^G12D^ Mice

*Ahr^+/+^* and *Ahr^−/−^* mice (C57BL/6N × 129 sV) were generated by homologous recombination in embryonic stem cells as described earlier [28] and used at the indicated experimental times. Conditional *K-Ras^G12D/+^* (*LSL-K-Ras^G12D/+^*) mice heterozygous for the G12D mutation were produced as reported [7] (Appendix A). *K-Ras^G12D/+^* mice were crossed with *Ahr^+/+^* and *Ahr^−/−^* mice to generate the *K-Ras^G12D/+^*;*Ahr^+/+^* and *K-Ras^G12D/+^*;*Ahr^−/−^* double mutant colonies. *K-Ras^G12D^* mice, in each Ahr genetic background, were maintained in heterozygosis since homozygous *K-Ras^G12D/G12D^* mice are lethal. *Ahr* and *K-Ras^G12D^* genotypes were analyzed by PCR using the oligonucleotides indicated in Appendix A. Experiments using mice were performed in accordance with the National and European legislation (Spanish Royal Decree RD53/2013 and EU Directive 86/609/CEE as modified by 2003/65/CE, respectively) for the protection of animals used for research. Experiments using mice were approved by the Bioethics Committee for Animal Experimentation of the University of Extremadura (Registry 109/2014) and by the Junta de Extremadura (EXP-20160506-1). Mice had free access to water and rodent chow. The animal research complied with the 3Rs strategy for replacement, reduction and refinement in the use of experimental animals.

### 2.2. K-Ras^G12D/+^ Activation in the Lung

To induce NSCLC in the lung, *K-Ras^G12D/+^*;*Ahr^+/+^* and *K-Ras^G12D/+^*;*Ahr^−/−^* mice were treated with the Ad5CMVCre adenovirus (Iowa University) by intranasal administration in order to delete the Stop signal included in the LSL cassette of the transgene, thus allowing the expression of the *K-Ras^G12D^* allele in the lung epithelium (Appendix A). Briefly, mice were anesthetized by i.p. injection of a mixture of ketamine:valium:atropine (0.25:2.0:0.1, mg/mL). Next, the adenoviral Ad5CMVCre suspension at 2.5 × 10^7^ pfu in 50 μL MEM medium was carefully dropped into each nostril of the nose using a micropipette equipped with a yellow tip. Mice were afterwards allowed to recover from treatment.

### 2.3. Lung Organoids

Lung tissues from AdenoCre-induced *K-Ras^G12D/+^*;*Ahr^+/+^* and *K-Ras^G12D/+^*;*Ahr^−/−^* mice were sectioned into pieces of 4 mm diameter and washed with ice-cold phosphate buffered saline (PBS) to remove traces of blood. Lung pieces were then dissociated into single cell suspensions by incubation in PBS-containing dispase (0.6 U/mL) (Roche, Basel, Switzerland), type-IV collagenase (60 U/mL) (Worthington, NJ, USA) and 0.0001% DNase solution (Roche) for 1 h at 37 °C with gentle shaking. Once digested, tissues were homogenized through a 21-gauge syringe, filtered by a 0.40 μm mesh and centrifuged at 300× *g* for 5 min. Pellets were resuspended and incubated for 3 min in RBC lysis buffer (Alfa Aesar-Thermo Fisher, Waltham, MA, USA) to lyse red blood cells. Next, cells were centrifuged and resuspended in PBS containing 0.5% BSA and 2 mM EDTA. Stem cell isolation was performed using the Pluripotent Stem Cell Isolation Kit (Miltenyi-Biotec, Bergisch, Germany) following the manufacturer’s recommendations. The same number of pluripotent stem cells were counted (200.000 cell/well), resuspended in a volume of MTEC/Plus medium (DMEM-F12 supplemented with 2.2 mg/mL sodium bicarbonate, 0.5% antibiotic/antimycotic, 0.5% insulin/transferrin, 0.1 μg/mL cholera toxin, 30 μg/mL epidermal growth factor, 30 μg/mL bovine pituitary extract, 5% FBS and 0.01 μM retinoic acid) containing growth factor-reduced Matrigel (1:1) (Corning, NY, USA) and plated at 37 °C and 5% CO_2_ in transwell inserts (Corning, NY, USA) covered with MTEC/Plus medium plus 2% ROCK Inhibitor (Sigma-Aldrich, San Louis, MO, USA). The next day, a fresh medium without ROCK inhibitor was added. Growth was monitored weekly for the presence of initiated organoids, which were maintained without passaging for no longer than four weeks. Thus, experiments were done with organoids of each genotype at passage one.

### 2.4. RNAseq and Bioinformatics

Total RNA from both AdenoCre-induced *K-Ras^G12D/+^*;*Ahr^+/+^* and *K-Ras^G12D/+^*;*Ahr^−/−^* lung tumors and healthy tissue was isolated using Trizol Reagent (Life Technologies, Carlsbad, CA, USA) and purified with the High Pure RNA isolation kit following the manufacturer’s instructions (Roche, Basel, Switzerland). RNA was quantified using Nanodrop equipment (Thermo-Fisher, Waltham, MA, USA) and its quality determined in an Agilent 2100 Bioanalyzer system. Only samples RNA integrity numbers (RIN) higher than 8.0 were used for analysis. RNAseq was performed using the Illumina technology, and at least five million paired-end reads (2 × 75 bp) were obtained per sample. These reads were aligned to the *Mus musculus* genome (mm10) using BWA; RPKM were calculated using the Salmon software. An ad hoc MATLAB pipeline generated from negative binomial distribution was used for estimation of adjusted *p*-values for each gene between *K-Ras^G12D/+^*;*Ahr^+/+^* and *K-Ras^G12D/+^*;*Ahr^−/−^* samples and for the subsequent visualization of the results (heatmaps and volcano plots). DAVID webserver was used to retrieve significant Gene Ontology terms and pathways for the differentially expressed genes (Appendix A).

### 2.5. Gene Expression Analyses

Total RNA was isolated from lung organoids derived from AdenoCre-activated *K-Ras^G12D/+^*;*Ahr^+/+^* and *K-Ras^G12D/+^*;*Ahr^−/−^* mice using the Trizol reagent (Life Technologies, Carlsbad, CA, USA). Samples were centrifuged and the supernatants precipitated with isopropanol. After additional centrifugation, pellets were dissolved in DEPC-treated water. A High Pure RNA Isolation Kit (Roche, Basel, Switzerland) was then used for further RNA purification. To analyze mRNA expression by RT-qPCR, reverse transcription was performed using random priming and the iScript Reverse Transcription Super Mix (Bio-Rad, Hercules, CA, USA). Real-time PCR was used to quantify mRNA expression for *Oct4*, *Nanog*, *Sox2*, *Klf4* and *c-Myc* using SYBR^®^ Select Master Mix (Life Technologies, Carlsbad, CA, USA) in a Step One Thermal Cycler (Applied Biosystems, Waltham, MA, USA) essentially as indicated [29]. *Gadph* was used to normalize gene expression (ΔCt) and 2^−ΔΔCt^ to calculate changes in mRNA levels with respect to control conditions. The primer sequences used are those indicated in Appendix A.

### 2.6. Hematoxylin/Eosin Staining

Lungs from AdenoCre-induced *K-Ras^G12D/+^*;*Ahr^+/+^* and *K-Ras^G12D/+^*;*Ahr^−/−^* mice were sectioned at 3–5 μm and processed for H&E staining as described [30]. Lungs were fixed overnight at room temperature in buffered formalin and included in paraffin. Following deparaffination and rehydration, sections were incubated for 3 min with hematoxylin, washed with tap water and stained with eosin for 1 min. After dehydration, samples were mounted and observed in a NIKON TE2000U microscope using 4× (0.10 numeric aperture) and 10× (0.25 numeric aperture) objectives.

### 2.7. Immunofluorescence

Lungs from AdenoCre-induced *K-Ras^G12D/+^*;*Ahr^+/+^* and *K-Ras^G12D/+^*;*Ahr^−/−^* mice were fixed in buffered formalin, deparaffinated and rehydrated in PBS. Antigen unmasking was performed by incubation in citrate buffer at pH 6. Sections were washed in PBS containing 0.05% Triton X-100 (PBS-T) and non-specific epitopes blocked in PBS-T containing 0.2% gelatin and 3% BSA (PBS-T-G-B) for 1 h at room temperature. Sections were then incubated overnight at 4 °C with the following primary antibodies diluted in PBS-T-G-B: anti-SFTPC (Millipore 1:200, Temecula, CA, USA), anti-SCGB1A1/CCL10 (Novus Biologicals, 1:100, Abingdon, UK), anti-SOX2 (Novus Biologicals 1:150), anti-ALDH1A1 (Abcam 1:200), anti-EPCAM (Santa Cruz Biotechnology 1:200), anti-LGR5 (Origene 1:200) and anti-PORCN (Millipore 1:150). Following washing in PBS-T, sections were incubated for 1 h at room temperature with Alexa-488, Alexa-550 or Alexa-633-labeled secondary antibodies diluted in PBS-T-G-B. After additional washing, sections were dehydrated and mounted in PBS:glicerol (1:1). Visualization was done in an Olympus FV1000 confocal microscope. Fluorescence analysis was done using the FV10 software (Olympus, Shinjuku, Japan) and Image J software. DAPI was used to stain cell nuclei. At least three replicates were performed of each condition analyzed. For each replicate, four to six fields were measured at the microscope.

### 2.8. Statistical Analyses

Quantitative data are shown as mean ± SD. Comparison between experimental conditions was done using GraphPad Prism 6.0 software (GraphPad, San Diego, CA, USA). Student’s *t*-test was used to analyze differences between the two experimental groups and ANOVA for the analyses of three or more groups. The Mann-Whitney non-parametric statistical method was used to compare rank variations between independent groups (* *p* < 0.05; ** *p* < 0.01, *** *p* < 0.001, **** *p* < 0.0001).

## 3. Results

### 3.1. Loss of Ahr Exacerbates K-Ras^G12D^-Driven Lung Tumorigenesis

Based on the tumor suppressor potential of Ahr found in the lung and other tissues [19,20,25,31], we first investigated the effects of the genetic ablation of Ahr in *K-Ras*^G12D^-driven NSCLC formation. To this end, we crossed *Ahr**^−^*^/*−*^ and *Cre*-inducible *K-Ras*^G12D/+^ mice to generate the compound *K-Ras*^G12D/+^;*Ahr**^−^*^/*−*^ strain. Cohorts of these mice and *K-Ras*^G12D/+^ controls were subsequently infected by nasal instillation with Cre-encoding adenoviruses to induce the formation of NSCLC, as previously described [7]. Using this approach, we found that *K-Ras^G12D/+^*;*Ahr**^−^*^/*−*^ mice developed NSCLC with higher burdens (Figure 1A,B) and shorter latencies (Figure 1C) than their single *K-Ras^G12D/+^* counterpart controls, with lesions evident at 12 weeks of age. Histological analyses of lung sections from those animals confirmed the increase in tumor burden and their earlier appearance with time (Figure 1B,C). In samples collected up to 25 weeks after Cre-mediated induction of *K-Ras*^G12D^, we found higher numbers of endobronchial hyperplasias in *K-Ras*^G12D/+^;*Ahr**^−^*^/*−*^ mice when compared to control *K-Ras*^G12D/+^ animals (Figure 1A, second and fourth panel from top). Likewise, we found higher numbers of type-II pneumocyte hyperplasias (reminiscent of adenocarcinomas) at 25 weeks post-induction than in controls (Figure 1A, second and fourth panel from top; Figure 1D). Finally, we also found higher percentages of papillary lesions arising from endobronchial hyperplasias in *K-Ras^G12D/+^*;*Ahr**^−^*^/*−*^ mice than in the control samples (Figure 1A, second and fourth panel from top; Figure 1E). Average tumor number per mouse and total tumor area were also increased in *K-Ras*^G12D/+^;*Ahr**^−^*^/*−*^ as compared to *K-Ras*^G12D/+^;*Ahr^+/+^* mice (Figure 1F,G). These data indicate that the loss of Ahr favors the initiation and progression of *K-Ras*^G12D^-driven NSCLC in mice.

### 3.2. The Ahr Deficiency Leads to Stem Cell Expansion in K-Ras^G12D^-Driven NSCLC

Type-II alveolar cells are considered lung stem cells with autorenewal capacity [32]. Immunofluorescence analyses showed that the number of SFTPC^+^ cells increased with time regardless of the genotype analyzed, although the levels remained higher in mice lacking Ahr (Figure 2A,B). This phenotype was most apparent at the latest time-points of these analyses (Figure 2B). Another subset of stem cells in the lung are the secretory Clara cells, located in bronchioles, which can be identified with the SCGB1A1 and CC10 markers [33]. The number of these cells was also markedly elevated in the case of *K-Ras^G12D/+^*;*Ahr^−/−^* mice when compared to control mice (Figure 2A,C). These results suggest that Ahr controls overall expansion and perpetuation of stem cells in the lung.

To dig dipper in this phenotype, we analyzed the expression of stem cell regulatory factors (SOX2, ALDH1A1, EPCAM, LGR5, PORCN) in the presence and absence of Ahr expression. SOX2 is a factor associated with the pluripotent state that has been shown to be important in the regulation of cell differentiation events in the epithelial component of the lung [34]. It also favors the proliferation of lung squamous cancer cells [35]. Aldehyde dehydrogenase (ALDH) has been associated with several types of normal and cancer stem cells, including those present in the lung [36]. EPCAM is another progenitor cell marker that has been proposed as a potential therapeutic target for lung adenocarcinoma [37]. LGR6 is a G-protein coupled receptor expressed in both stem cells and epithelial adenomas [38]. PORCN is an acetyltransferase, important for maintaining the cancer stem cell niche, that participates in the secretion of proliferative factors that activate LGR5 such as, for example, Wnt [39]. Using immunofluorescence studies, we found that the lung sections from *K-Ras^G12D/+^*;*Ahr^−/−^* mice contained larger numbers of SOX2^+^ (Figure 3 and Figure 4A), ALDH1A1^+^ (Figure 3 and Figure 4B), EPCAM^+^ (Figure 3 and Figure 4C), LGR5^+^ (Figure 5A,B) and PORCN^+^ (Figure 5A,C) cells than those from single *K-Ras^G12D/+^* mice at most time-points analyzed. Thus, in agreement with the increased numbers of SFTPC^+^ and SCGB1A1^+^/CC10^+^ stem cell populations, our findings indicate that the loss of Ahr leads to an increase in a large variety of stem cell-connected regulatory factors in the lung.

To investigate if Ahr had a causal role in the regulation of gene expression programs in *K-Ras*^G12D^-expressing cells, we performed RNA-seq analyses using two different comparative conditions: (i) *K-Ras^G12D/+^*;*Ahr^−/−^* versus *K-Ras*^G12D/+^;*Ahr^+/+^* in healthy lung tissue and (ii) *K-Ras^G12D/+^*;*Ahr^−/−^* versus *K-Ras*^G12D/+^;*Ahr^+/+^* in tumor tissue. In both cases, we found a significant number of genes whose expression was dependent on Ahr status (Figure 6A,B). The number of differentially regulated genes was, however, much larger in the case of tumor samples (730 genes) than in those from healthy tissue (224 genes) (Figure 6A,B). Volcano plots aimed to identify meaningful changes in gene expression also identified DEGs that were either upregulated or downregulated depending on the status of Ahr expression in the lungs (Figure 6C,D). Interestingly, and consistent with our previous observations (Figure 2, Figure 3, Figure 4 and Figure 5), we found that the samples from *K-Ras^G12D/+^*;*Ahr^−/−^* mice displayed a statistically significant upregulation of the stem cell state-related *Nanog*, *Klf4* and *Sox2* transcripts (Figure 6D). GO analysis identified several ontology terms related known to have a role in differentiation and tissue transformation (Appendix A).

### 3.3. Lack of Ahr Enhances 3D Lung Organoids Formation by K-Ras^G12D^

To provide further support for the effect of Ahr expression status on the oncogenic activity of *K-Ras*^G12D^ in the lung, we performed 3D organoid cultures to study biological features specifically associated with both epithelial cell undifferentiation and pluripotency. Organoids at passage one were used in all the experiments. We observed that the Ahr deficiency promoted the formation of increased numbers of organoids by *K-Ras*^G12D^ (Figure 7A,B). However, we did not find any statistically significant change in the size of those organoids when using cells from compound *K-Ras^G12D/+^*;*Ahr^−/−^* and single *K-Ras^G12D/+^* mice (Figure 7C). Despite this, we observed that the organoids derived from *K-Ras^G12D/+^*;*Ahr^−/−^* cells could form more differentiated bronchioalveolar structures at the longer culturing times (Figure 7A,D). This suggests that Ahr deficiency induces an undifferentiated status in *K-Ras*^G12D^-expressing cell precursors that, eventually, also favors their potential to differentiate into bronchioalveolar structures.

We also observed that the *K-Ras^G12D/+^*;*Ahr^−/−^* organoids displayed higher levels of mRNAs encoding pluripotency and undifferentiation markers such as *Oct4*, *Nanog*, *Sox2* and *Myc* than those generated from *K-Ras*^G12D^-expressing cells (Figure 8A–D). By contrast, the transcripts levels of *Klf4* were lower in the organoids lacking Ahr (Figure 8E). These expression patterns were probably autonomously driven by Ahr, given that the same pattern of transcript regulation was observed when using samples from *Ahr^−/−^* and *K-Ras^G12D/+^*;*Ahr^−/−^* organoids (Figure 8A–E).

## 4. Discussion

An increasing number of studies are revealing novel and relevant functions of Ahr in normal cell physiology and in diverse pathological conditions. Accordingly, it is well-known that Ahr has oncogenic or tumor suppressor activity in specific cell, tissue and organ contexts [18,19,20]. Regarding lung cancer, some studies support Ahr as a NSCLC promoter [24], whereas other works suggest its role as an inhibitor of NSCLC growth and dissemination [25,26]. Previous work from ours and from other laboratories have shown that Ahr deficiency promotes an undifferentiated and pluripotent phenotype in different cell types [40,41], eventually improving lung and liver regeneration against acute damage through the expansion of stem cell subsets [27,42]. Various oncogenes have also been found to have protumoral activity in the lung; among them, the G12D mutation in *K-Ras* appears associated with a bad outcome and multifocal NSCLC disease [13,14].

Therefore, the question remains of whether the Ahr interaction with oncogenic *K-Ras^G12D^* enhances or inhibits NSCLC. To investigate this, the conditional *K-Ras^G12D^* gene was expressed in an Ahr-deficient mouse strain and the combined genotype analyzed for its susceptibility to NSCLC formation. Indeed, *K-Ras^G12D/+^*;*Ahr^−/−^* mice were more susceptible to NSCLC than their Ahr-expressing counterparts, implying a tumor suppressor role for Ahr. Within the time window analyzed, Ahr-deficient lesions were enriched in endobronchial hyperplasias, type-II pneumocyte hyperplasias (indicative of adenocarcinomas) and endobronchial papillary lesions. Importantly, the latency of such NSCLC lesions in *K-Ras^G12D/+^*;*Ahr^−/−^* mice was significantly shorter than in *K-Ras^G12D/+^*;*Ahr^+/+^* mice, suggesting that Ahr deficiency stresses the tumorigenic potential of *K-Ras^G12D^* in the lung.

It is possible that Ahr restrains NSCLC by controlling the expansion of lung stem cells, which could eventually counteract the protumoral activity of *K-Ras^G12D^*. Accordingly, major lung stem cells responsible for regeneration and repair after injury, including type-II alveolar cells and Clara cells, were amplified in *K-Ras^G12D/+^*;*Ahr^−/−^* NSCLC lesions. Further, cells positive for pluripotency, stemness and cancer stem cell niche markers SOX2, ALDH1A1, EPCAM, LGR5 and PORCN were also significantly expanded in Ahr-deficient lesions along the time window analyzed. Notably, all these cell subsets were already amplified in *K-Ras^G12D/+^*;*Ahr^−/−^* mice at the very early stages of the NSCLC lesions, suggesting that constitutive Ahr deficiency may prime the lung for *K-Ras^G12D^*-dependent tumorigenesis. In agreement, we previously found that Ahr-null mice are more efficient at repairing lung injury by expanding Clara and Basal stem cell subpopulations [27]. Moreover, RNAseq revealed the existence of differential gene expression patterns and confirmed the upregulation of cells expressing pluripotency genes *Sox2*, *Nanog* and *Klf4* in *K-Ras^G12D/+^*;*Ahr^−/−^* versus *K-Ras^G12D/+^*;*Ahr^+/+^* NSCLCs.

Tissue-derived 3D organoids have emerged as a relevant ex vivo model to study stem cell self-organization and differentiation into specific cell types [43]. Based on our previous data, we decided to establish organoid cultures from *K-Ras^G12D/+^*;*Ahr^−/−^* and *K-Ras^G12D/+^*;*Ahr^+/+^* lungs. Supporting our hypothesis, lack of Ahr not only increased the number of 3D structures formed by *K-Ras^G12D/+^*;*Ahr^−/−^* lung epithelial cells but also improved their efficiency to differentiate into bronchioalveolar structures. Strengthening our hypothesis, gene expression analyses revealed the overexpression of *Oct4*, *Nanog*, *Sox2* and *c-Myc*, but not *Klf4* in *K-Ras^G12D/+^*;*Ahr^−/−^*-derived organoids. It is likely that these expression patterns are Ahr-driven since the same profile of transcript regulation was found in *Ahr^−/−^* lung epithelial cells expressing endogenous levels of normal K-Ras.

In summary, we report here that *Ahr* counteracts the protumoral activity of *K-Ras^G12D^* in NSCLC, probably by controlling the expansion of subsets of lung stem cells. Consistently, the oncogenic potential of *K-Ras^G12D^* is exacerbated in the absence of Ahr to increase the NSCLC burden. Thus, Ahr has tumor suppressor activity under oncogenic *K-Ras^G12D^* activation in NSCLC. Upregulation of Ahr by non-toxic physiological agonists may represent an opportunity to reduce the burden and severity of lung tumorigenesis caused by the *K-Ras^G12D^* mutation in humans.

## 5. Conclusions

The aryl hydrocarbon receptor Ahr counteracts the oncogenic potential of *K-Ras^G12D^* in the lung, thus suppressing non-small cell lung cancer. Such tumor suppressor activity takes place by controlling the expansion of lung stem cells and by limiting the expression of pluripotency-inducing genes. Physiological modulation of Ahr activity by non-toxic receptor ligands may have clinical interest in oncogene-dependent NSCLC.

## Figures and Tables

**Figure 1 cancers-13-04071-f001:**
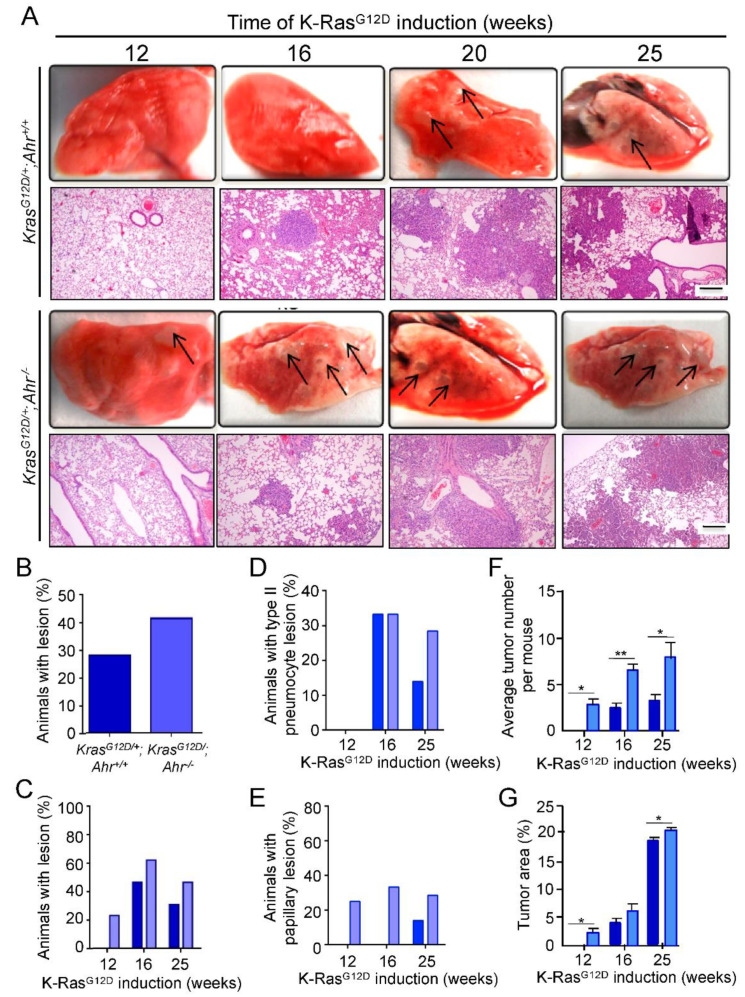
Lung lesions induced by *K-Ras^G12D^* are modulated by the *Ahr* genotype. (**A**) *K-Ras^G12D/+^*;*Ahr^+/+^* and *K-Ras^G12D/+^*;*Ahr*^−/−^ mice were treated with AdenoCre and the lung lesions generated analyzed for up to 25 weeks macroscopically (upper panels) and histologically by H&E staining (lower panels. (**B**) Histological quantification of the percentage of mice of each genotype developing NSCLC lesions. (**C**) Histological quantification of the percentage of mice with NSCLC lesions upon the time of *K-Ras^G12D^* activation at up to 25 weeks. (**D**,**E**) Percentage of mice developing type-II pneumocyte hyperplasia (**D**) or papillary lesions (**E**). (**F**,**G**) Quantification of overall tumor burden (**F**) and the total tumoral area comprising endobronchial hyperplasia, type-II pneumocyte hyperplasia (reminiscent of adenocarcinoma) and papillary lesions were measured using Fiji ImageJ software (**G**). Groups of 4–5 mice were used for each genotype and time point. Tumor incidence is represented as the percentage of affected mice with respect to the total number of mice induced. Bar corresponds to 100 μm. Data are shown as mean + SD (* *p* < 0.05; ** *p* < 0.01).

**Figure 2 cancers-13-04071-f002:**
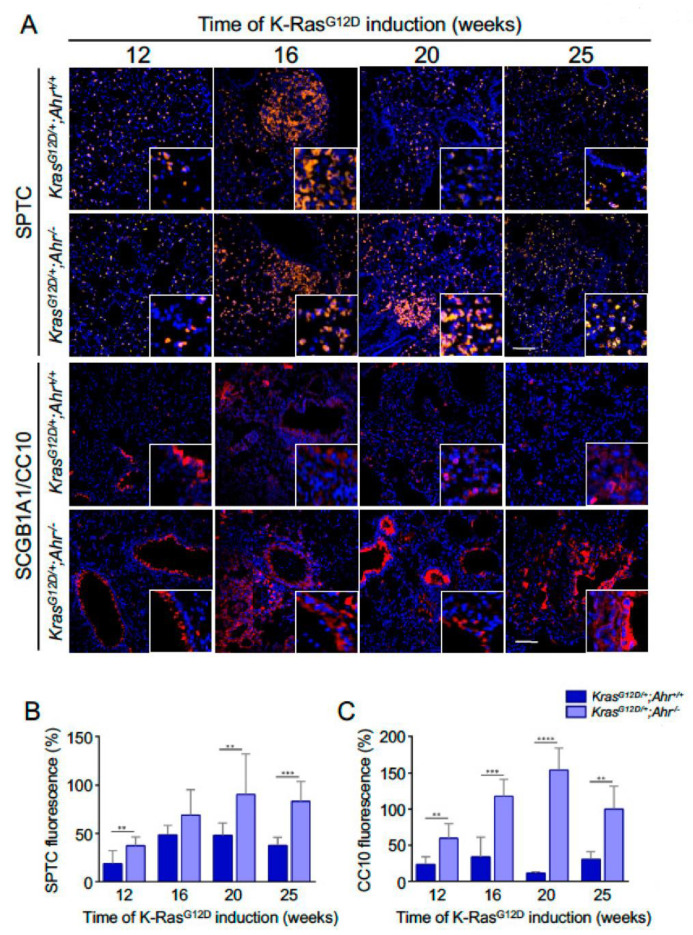
Lung progenitor cells expressing SFTPC and SCGB1A1/CC10 markers are upregulated in *K-Ras^G12D/+^*;*Ahr*^–/–^ mice**.** (**A**) Sections from *K-Ras^G12D/+^*;*Ahr^+/+^* and *K-Ras^G12D/+^*;*Ahr*^−/−^ lungs were analyzed by immunofluorescence to identify surfactant-producing (SFTPC^+^) or Clara (SCGB1A1/CC10^+^) progenitor cells. (**B**,**C**) The expansion of SFTPC-positive (**B**) or SCGB1A1/CC10-positive (**C**) cells with time of *K-Ras^G12D^* induction was quantified from 12 to 25 weeks in each mouse genotype. The conjugated secondary antibodies were Alexa-550 and Alexa-633 for SFTPC and SCGB1A1, respectively. Sections were analyzed using an Olympus FV1000 confocal microscope and the FV10 software (Olympus, Shinjuku, Japan). DAPI staining was used to label cell nuclei. Bar corresponds to 100 μm. Data are shown as mean + SD (** *p* < 0.01; *** *p* < 0.001; **** *p* < 0.0001).

**Figure 3 cancers-13-04071-f003:**
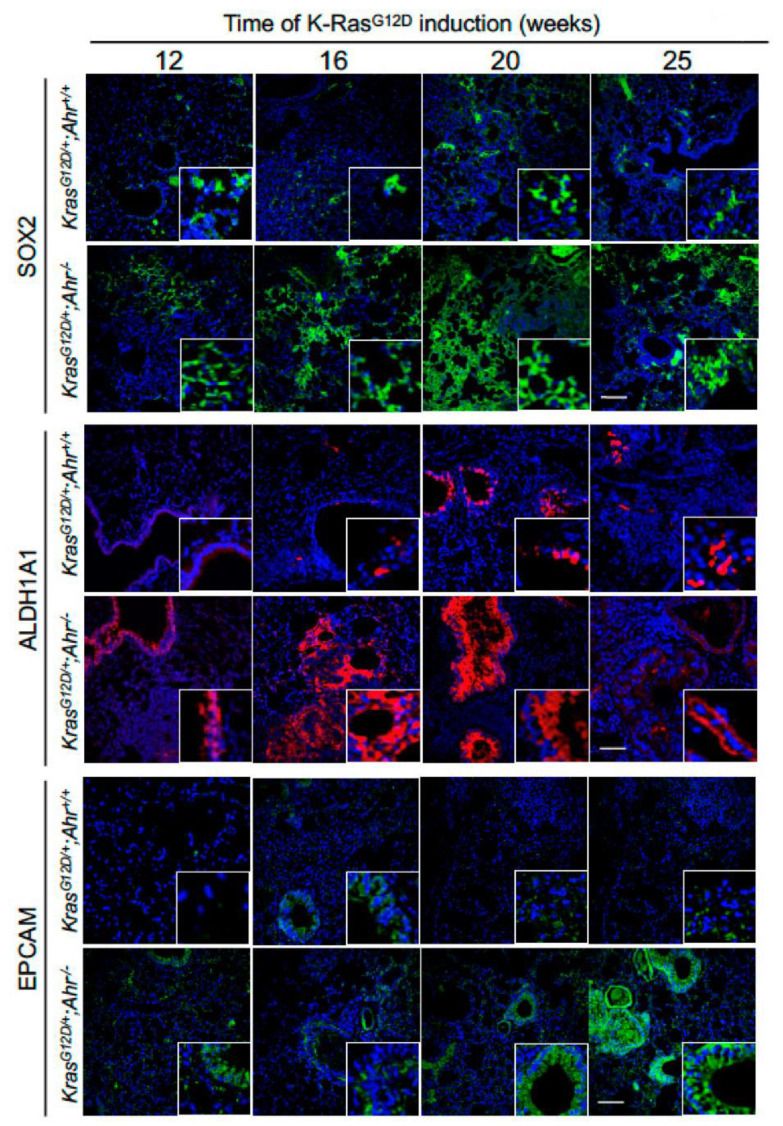
Lung progenitor cells expressing tumor-associated SOX2, ALDH1A1 and EPCAM markers are upregulated in *K-Ras^G12D/+^*;*Ahr*^−/−^ mice. Lung sections from *K-Ras^G12D/+^*;*Ahr^+/+^* and *K-Ras^G12D/+^*;*Ahr*^−/−^ mice were processed and analyzed by immunofluorescence using specific antibodies for SOX2 (upper), ALDH1A1 (middle) and EPCAM (lower) for up to 25 weeks of *K-Ras^G12D^* activation. Conjugated secondary antibodies were Alexa-488 for SOX2 and EPCAM and Alexa-633 for ALDH1A1. Sections were analyzed using an Olympus FV1000 confocal microscope and the FV10 software (Olympus, Shinjuku, Japan). DAPI staining was used to label cell nuclei. Bar corresponds to 100 μm.

**Figure 4 cancers-13-04071-f004:**
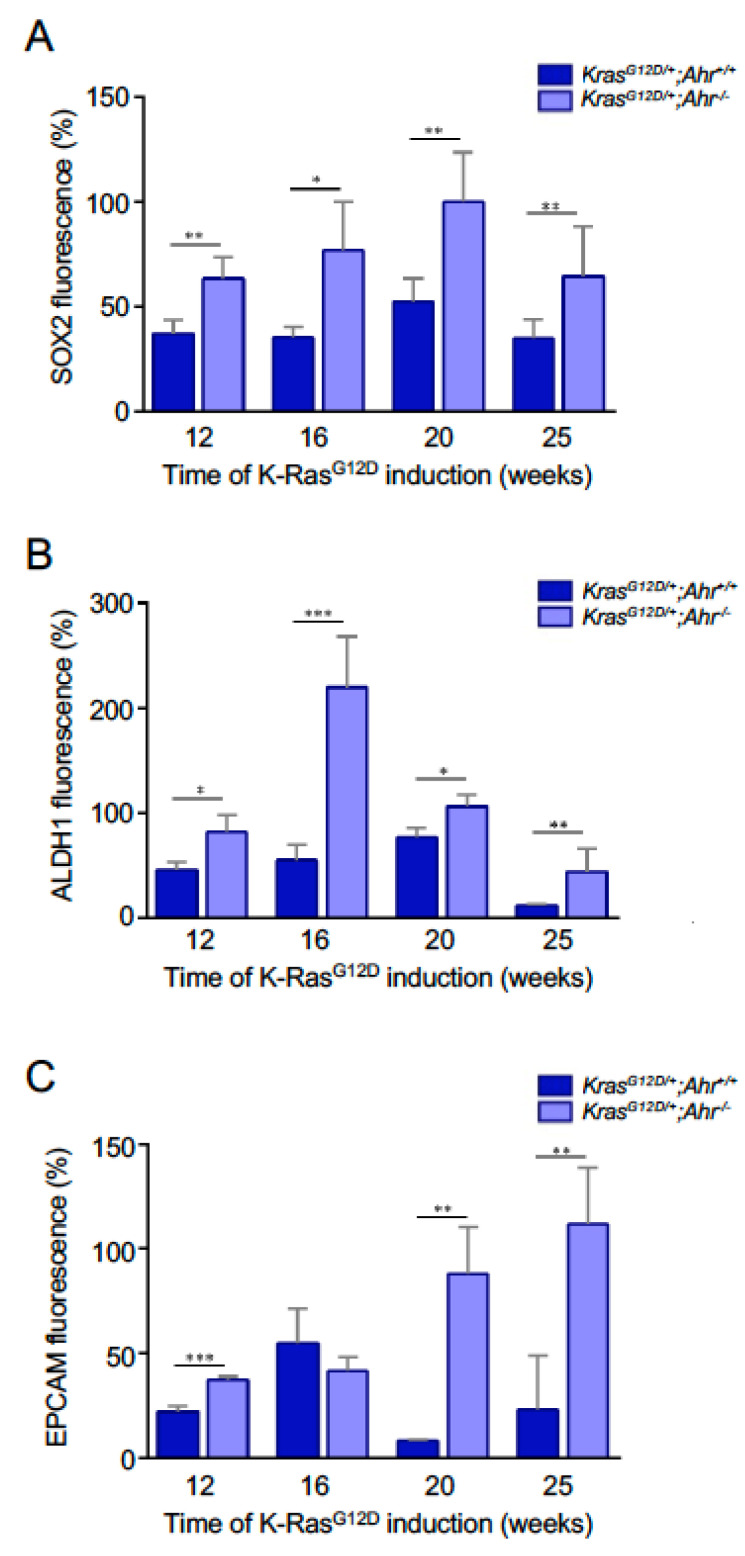
Quantification of lung progenitor cells expressing tumor-associated SOX2, ALDH1A1 and EPCAM. Lung sections from *K-Ras^G12D/+^*;*Ahr^+/+^* and *K-Ras^G12D/+^*;*Ahr*^−/−^ mice (see Figure 3) were quantified for the presence of SOX2 (**A**), ALDH1A1 (**B**) and EPCAM (**C**)-positive cells during *K-Ras^G12D^* induction from 12 to 25 weeks. Data are shown as mean + SD (* *p* < 0.05; ** *p* < 0.01; *** *p* < 0.001). Number of replicates (*N* = 3) and measures taken from 4–6 fields for each replicate.

**Figure 5 cancers-13-04071-f005:**
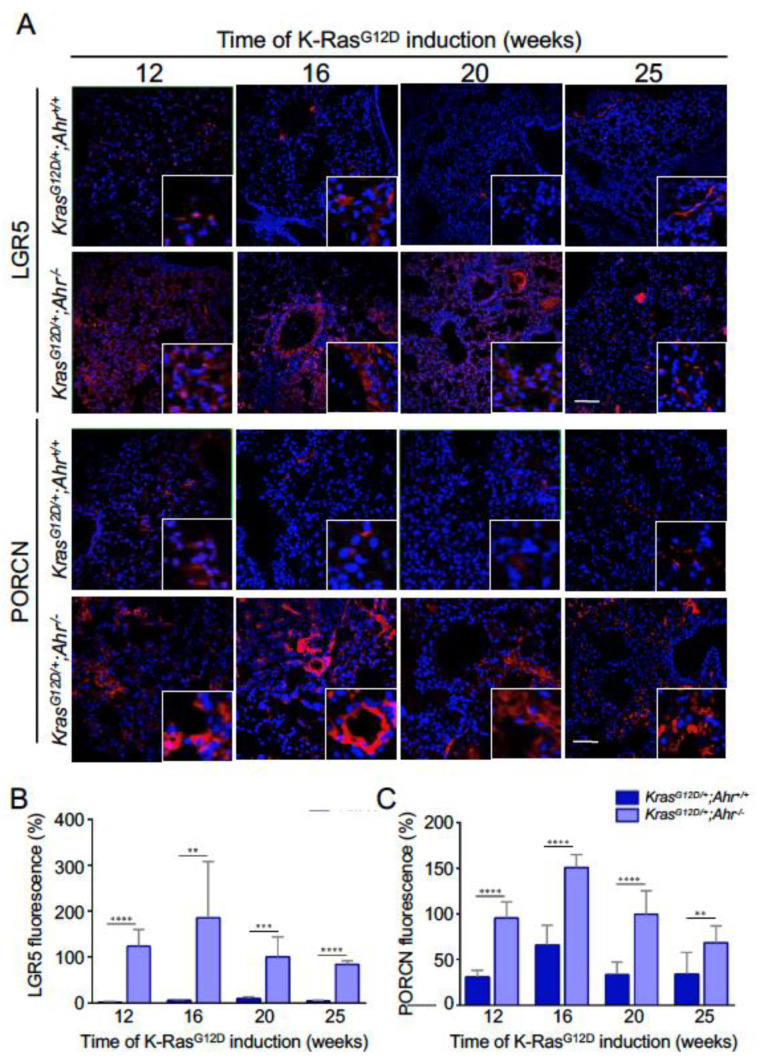
Markers of the cancer stem cell niche LGR5 and PORCN are overrepresented in Ahr-null K-RasG12D mice. (**A**) Sections from *K-Ras^G12D/+^*;*Ahr^+/+^* and *K-Ras^G12D/+^*;*Ahr*^−/−^ lungs were analyzed by immunofluorescence to identify cancer stem cells expressing LGR5 (upper) or PORCN (lower) as surrogate markers for tumor progression. (**B**,**C**) The percentages of LGR5-positive (**B**) or PORCN-positive cells (**C**) after *K-Ras^G12D^* activation from 12 to 25 weeks were quantified. The conjugated secondary antibody was Alexa-633 for both markers. Sections were analyzed using an Olympus FV1000 confocal microscope and the FV10 software (Olympus, Shinjuku, Japan). DAPI staining was used to label cell nuclei. Bar corresponds to 100 μm. Data are shown as mean + SD (** *p* < 0.01; *** *p* < 0.001; **** *p* < 0.0001).

**Figure 6 cancers-13-04071-f006:**
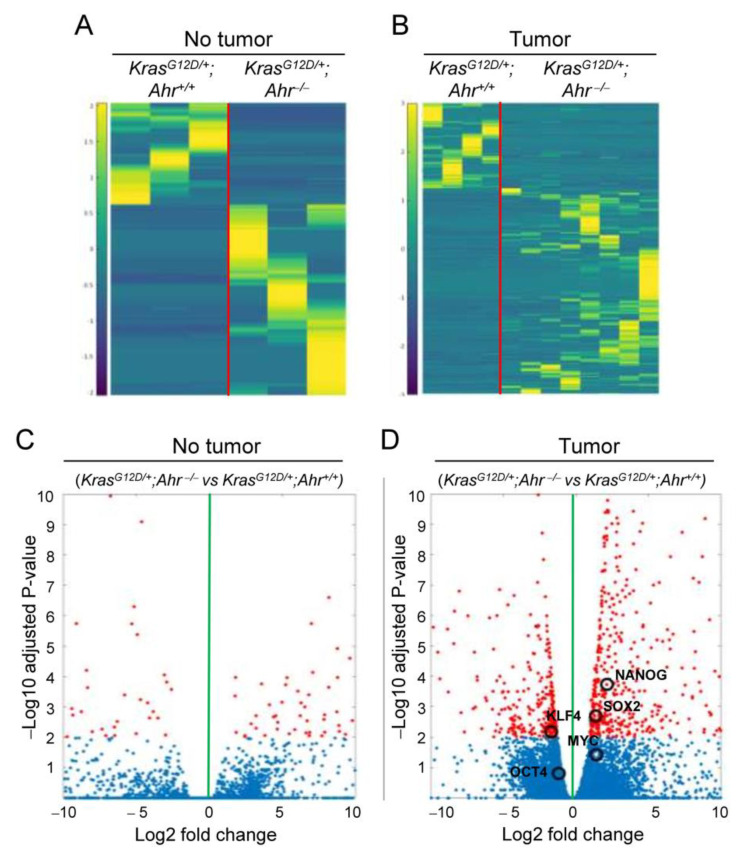
The Ahr genotype modulates gene expression patterns in normal and K-RasG12D-expressing NSCLC tumors. Non-tumor and tumor tissues were recovered from *K-Ras^G12D/+^*;*Ahr^+/+^* and *K-Ras^G12D/+^*;*Ahr*^−/−^ lungs and analyzed for global gene expression by RNAseq. (**A**,**B**) Heatmaps of differentially expressed genes (DEG) in non-tumoral (**A**) and tumoral lung tissue (**B**) of *K-Ras^G12D/+^*;*Ahr*^−/−^ mice as normalized by the expression levels present in *K-Ras^G12D/+^*;*Ahr^+/+^* mice. (**C**,**D**) Volcano plots of non-tumoral (**C**) and tumoral lung tissue (**D**) DEGs of *K-Ras^G12D/+^*;*Ahr*^−/−^ mice as normalized by their expression in *K-Ras^G12D/+^*;*Ahr^+/+^* mice. Samples showing significant DEGs between *K-Ras^G12D/+^*;*Ahr^+/+^* and *K-Ras^G12D/+^*;*Ahr*^−/−^ genotypes (adjusted *p*-value < 0.01 and absolute Log_2_ fold change > 1) are showed in red dots; non-DEGs are shown in blue dots. The expression of pluripotency genes *Nanog* (*p* < 0.001), *Sox2* (*p* = 0.002), *Klf4* (*p* = 0.004), *Oct4* (*p* = 0.4) and *c-Myc* (*p* = 0.12) are highlighted within open circles. *N* = 4 for each condition assayed.

**Figure 7 cancers-13-04071-f007:**
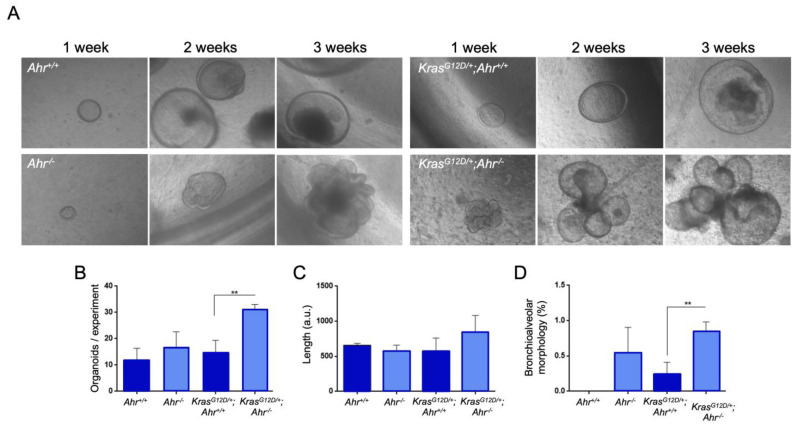
Ahr deficiency favors the formation of lung organoids and their differentiation to bronchioalveolar structures upon *K-Ras^G12D^* activation. (**A**) *K-Ras^G12D/+^*;*Ahr^+/+^* and *K-Ras^G12D/+^*;*Ahr**^−^*^/*−*^ lung cells were cultured under ex vivo conditions for up to three weeks to generate organoids. (**B**,**C**) The number of organoids generated from 2 × 10^5^ initiating cells of each genotype were quantified (**B**) and their length measured (**C**). (**D**) Organoids were analyzed for their bronchioalveolar morphology and the resulting numbers plotted at three weeks after culturing. Data are shown as mean + SD (** *p* < 0.01). Number of organoids (*N* = 4) and replicates of each experiment (*n* = 3–6). Organoids at passage one were used in all the experiments.

**Figure 8 cancers-13-04071-f008:**
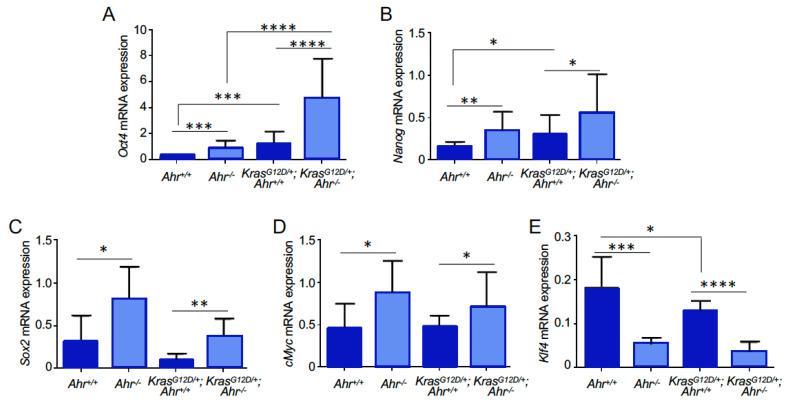
Representative pluripotency genes are overexpressed in Ahr-deficient organoids. (**A**–**E**) *K-Ras^G12D/+^*;*Ahr^+/+^*, *K-Ras^G12D/+^*;*Ahr*^−/−^, *Ahr^+/+^* and *Ahr*^−/−^ organoids from lung cells were analyzed for the mRNA expression of *Oct4* (**A**), *Nanog* (**B**), *Sox2* (**C**), *c-Myc* (**D**) and *Klf4* (**E**) by RT-qPCR. Data are shown as mean + SD (* *p* < 0.05; ** *p* < 0.01; *** *p* < 0.001 and **** *p* < 0.0001). Number of organoids (*N* = 3) and replicates of each experiment (*n* = 2–4).

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
