# Peer review of "Loss of Aryl Hydrocarbon Receptor Favors K-RasG12D-Driven Non-Small Cell Lung Cancer"

_cancers, 2021, doi:10.3390/cancers13164071_

Round 1

Reviewer 1 Report

The authors have addressed most of the concerns raised in the original review. However, the response to the reviewer query "In the case of lung organoids, the authors should specify the generation number of the organoid used for their experiments" is complete. While the authors have provided the number of organoids used and the number of starting cells, they have not provided the 'generation number' (passage number) of the organoids that  were used for the experiments. Where they from passage 1, 2, 4, or 4?  The authors should clarify and state it in the 'Materials and Methods' section and in the respective Figure legend.

Author Response

We thank the reviewer for their positive evaluation of our manuscript. Attending to his/her current query, we have now added in the Methods section (page 3, lines 149, 150), Results (page 13, line 366) and in the legend for Figure 7 (page 13, lines 382, 383) that the experiments were performed using organoids of each genotype at passage 1.

Reviewer 2 Report

The authors have addressed my concerns. I am happy to recommend the publication of the manuscript.

Author Response

We thank this reviewer for his/her positive consideration to our manuscript and for the recommendation made to the editorial office. We also acknowledge his review and attention to our work.